# Bone Marrow Stromal Cells Drive Key Hallmarks of B Cell Malignancies

**DOI:** 10.3390/ijms21041466

**Published:** 2020-02-21

**Authors:** Maurizio Mangolini, Ingo Ringshausen

**Affiliations:** 1Wellcome Trust/MRC Cambridge Stem Cell Institute, University of Cambridge, Cambridge CB2 0AH, UK; mm2084@medschl.cam.ac.uk; 2Department of Haematology, Addenbrooke’s Hospital, Cambridge University hospital, Cambridge CB2 0AH, UK

**Keywords:** mesenchymal cells, bone marrow stroma, lymphoma, CLL

## Abstract

All B cell leukaemias and a substantial fraction of lymphomas display a natural niche residency in the bone marrow. While the bone marrow compartment may only be one of several sites of disease manifestations, the strong clinical significance of minimal residual disease (MRD) in the bone marrow strongly suggests that privileged niches exist in this anatomical site favouring central elements of malignant transformation. Here, the co-existence of two hierarchical systems, originating from haematopoietic and mesenchymal stem cells, has extensively been characterised with regard to regulation of the former (blood production) by the latter. How these two systems cooperate under pathological conditions is far less understood and is the focus of many current investigations. More recent single-cell sequencing techniques have now identified an unappreciated cellular heterogeneity of the bone marrow microenvironment. How each of these cell subtypes interact with each other and regulate normal and malignant haematopoiesis remains to be investigated. Here we review the evidences of how bone marrow stroma cells and malignant B cells reciprocally interact. Evidently from published data, these cell–cell interactions induce profound changes in signalling, gene expression and metabolic adaptations. While the past research has largely focussed on understanding changes imposed by stroma- on tumour cells, it is now clear that tumour-cell contact also has fundamental ramifications for the biology of stroma cells. Their careful characterisations are not only interesting from a scientific biological viewpoint but also relevant to clinical practice: Since tumour cells heavily depend on stroma cells for cell survival, proliferation and dissemination, interference with bone marrow stroma–tumour interactions bear therapeutic potential. The molecular characterisation of tumour–stroma interactions can identify new vulnerabilities, which could be therapeutically exploited.

## 1. Introduction

In the past 20 years, we have witnessed how technical advances in sequencing technologies have informed us about the genetic abnormalities underlying many B cell malignancies and, based on bulk sequencing studies, recurrent and rare mutations have been identified, allowing further sub-classifications of these diseases. Through deep-sequencing and mathematical modelling, driver mutations can now be distinguished from sub-clonal passenger mutations, present in only a fraction of cells, and it is expected that single-cell technologies will further inform us about clonal and sub-clonal events (genetic mutations, epigenetic alterations and differential protein expression) occurring in an individual cell. There is, however, a discrepancy in the translation of this knowledge into targeted therapies, which is significantly trailing behind as most patients are still treated with combinations of monoclonal antibodies and conventional chemotherapies, such as CHOP (cyclophosphamide, hydroxydaunorubicin, vincristine sulfate (Oncovin), and prednisone) regimen, Bendamustine or Chlorambucil. The more recent introduction of targeted therapies, antagonising central signalling nodes in the B cell receptor pathway or BH3-only proapoptotic proteins, has further advanced the spectrum of therapeutics and demonstrated impressive clinical responses in some patients; however, the dogma of “indolent lymphoma” equals “inability to cure” still remains accurate. Although treatments are highly effective for many patients, a large fraction of patients inevitably relapse months and years following treatment. The key biological processes underlying this tumour-cell dormancy are largely unknown. Clinically, residual tumour cells that survive therapy are classified as minimal-residual disease (MRD), whereby the methods used to identify these cells vary across patients and diseases, depending on the availability of technologies and the invasiveness of the clinical procedure (e.g., biopsy, PET-scan). In this regard, the bone marrow compartment is so easily accessible that even non-surgeons can perform the procedure, and therefore, most of our knowledge about the underlying cellular and molecular mechanisms driving MRD originate from investigations of this particular compartment.

## 2. Cellular Heterogeneity of Bone Marrow Stroma Cells

The niche requirements for tumour cell dormancy have not been described, and it remains largely unknown in what tissues they are present, as diagnostic procedures to assess MRD-status are restricted to easily accessible tissues. Attributed to these circumstances, in haematological malignancies, the bone marrow is the best studied localisation where residual tumour cells can be detected with minimally invasive techniques. It is, therefore, logical to assume that resident stromal cells provide signals for tumour cells, allowing them to survive cytotoxic therapies. It is tempting to assume that other protective niches in different organs of the human body must exist where tumour cells find conditions allowing them to withstand cytotoxic therapies. However, a strong argument against this assumption is the relatively high predictive value of the bone marrow MRD status for disease recurrence, indicating that this anatomical side is more specialised than other tissues to shelter tumour cells from cytotoxic agents. Biologically, this may be based on the fact that this compartment is the natural home for haematopoietic cells. Alternatively, the bone marrow MRD status may merely be a good indicator for residual disease also in other organs.

The bone marrow is a complex organ and harbours at least two hierarchically organised systems with the main purpose to produce and release more than 200 million blood cells every day in the bloodstream [1]. Central to its functions are hematopoietic stem cells (HSCs), essential in providing mature, differentiated cells while maintaining their stemness throughout the life-span of the organism. The state of HSCs, differentiation versus self-renewal versus quiescence is exclusively dependent on and regulated by non-hematopoietic cells, generally referred to as stromal cells.

Notably, HSCs are not the only cells in the bone marrow with stemness properties: bone marrow mesenchymal stem cells (BM-MSCs) are characterised by ‘stem-like’ properties allowing them to differentiate into different lineages, including adipocytes, chondrocytes, and osteoblasts in order to form bones, cartilage, bone marrow, fat and ‘hematopoietic-supporting’ tissues [2]. The drive to study bone marrow mesenchymal stroma cells (BMSCs) originates from the observations that they essentially can control all aspects of HSC-biology, therefore making it almost impossible to understand the mechanisms of blood production without defining the role and contribution of BMSCs. While this topic is undoubtedly interesting, it is outside the focus of this review, and we may refer the reader to some of the many excellent reviews covering this subject [3,4,5].

In 2006, the International Society for Cellular Therapy (ISCT) described the minimal characteristics of BMSCs as: (1) adherence to cell culture plastic under standard conditions, (2) expression of the surface markers CD105, CD73, and CD90 and absence of others (CD45, CD34, CD11b or CD14, CD79α or CD19, and HLA-DR) and (3) lineage differentiation capacity into osteoblasts, chondrocytes and adipocytes [6]. Naturally, over the years, this classification has been further refined, now moving towards a single-cell characterisation of the different stromal populations in in vivo models. Remarkably, a recent study using single-cell RNA sequencing (scRNA-seq) identified six different cell types of stromal cells with 17 distinct subsets, providing a comprehensive gene expression census of the BMSCs in adult mice [7]. These RNA fingerprints distinguished MSCs, MSC-descendent osteolineage cells (OLCs), chondrocytes, fibroblasts, endothelial cells and pericytes and identified their lineage trajectories, highlighting the limitations of previously used marker-based classifications. Importantly, scRNA-seq allowed for the first time to distinguish fibroblasts from MSCs and pericytes. This study was further advanced by another report published by the same group. In this, using mass cytometry (CyTOF)-based single-cell protein analysis, the authors show that under stress conditions, such as irradiation, only 3 out of 28 bone marrow stromal subsets previously identified in homeostatic conditions persisted, suggesting a deep remodelling of the stroma microenvironment following a genotoxic insult. Surprisingly, the authors identified that some stromal cell populations, traditionally considered to be relevant for haematopoiesis and including the LeptinR^+^ and Nestin^+^ subsets, were lost [8]. In this study, the additional analysis performed under stress conditions is particularly significant as it allows to identify cell subsets that are most likely involved in more common clinical stress conditions, such as early stages of hematopoietic regeneration following chemotherapy. Evidently, any mammalian organism needs to be able to adapt its hematopoietic system to stress circumstances, imposed by infections, blood loss, etc. These changes are executed through signalling changes in either cell compartment.

Cancer and chemotherapy constitute their own kind of stress to the bone marrow and, expectedly, have profound effects on the composition and functions of bone marrow stromal cells. This was shown recently by the Aifantis group through applying scRNA seq from marker-defined (VE-cadherin, leptin-receptor and collagen 2.3) subpopulations of bone marrow stromal cells. They identified a so-far unexpected cellular heterogeneity and a transcriptional reprogramming of stroma cells induced by chemotherapy [9]. This is exemplified by an enrichment in adipocytes under stress conditions, associated with a down-regulation of NOTCH ligands Delta Like Canonical Notch Ligand 1 (DLL1) and DLL4 in the perivascular niche, skewing cell production towards the myeloid lineage [9].

A similar analysis, performed in an Acute Myeloid Leukaemia (AML) mouse model, demonstrated a disease-induced distortion of the stromal compartment with a predominant shift in MSC-descendent osteolineage cells (OLC)-cells, causing a loss of the bone maturation phenotype. Furthermore, this study shows that leukemic cells de-regulate the production of HSC-niche factors in four out of six major stroma subpopulations, which likely contributes to bone marrow failure observed in almost every AML patient [7]. However, it remains to be shown to what extent these findings apply to lymphatic malignancies.

The data illustrate how advances in sequencing techniques and data analysis have remarkably improved our knowledge of the stromal cell composition in the mouse bone marrow. How relevant these findings are for patients remains to be determined.

## 3. Effects of BMSCs on Malignant B Cells

A lack of unique markers or promoters, which could be used to reliably identify and genetically modify all of the newly described BMSC subsets remains a major hurdle in studying the function of stromal cells in the context of B cell malignancies in vivo. Therefore, the overwhelming majority of data originates from in vitro experiments, employing co-culture systems of malignant B cells and stromal cells, assuming that findings can be extrapolated to in vivo scenarios. The importance of stromal cells for lymphoid malignancies was initially highlighted by the observation that the majority of primary transformed B cells rapidly die ex vivo but are rescued from apoptosis when cultured in direct contact with BMSCs [10,11] (Figure 1). Additionally, these protective effects can be extended to stress conditions originating from cytotoxic agents, demonstrating that stromal cells can protect primary leukemic cells not only from spontaneous but also from ‘drug-induced’ apoptosis [12,13]. It is reasonable to speculate that the very same protective mechanisms contribute to MRD; hence, their understanding may allow the development of therapies that enable to fully eradicate malignant B cells.

### 3.1. BMSCs and Apoptosis

In order to fulfil this environment-mediated anti-apoptotic activity, BMSCs need to be able to trigger anti-apoptotic signals in leukemic cells. This is exemplified in the BMSC-mediated up-regulation of the BCL-2 family of anti-apoptotic proteins in malignant B cells via direct cell–cell contact or soluble factors secreted by BMSCs. Evidence for the importance of BCL-2 dependent survival induced by stromal cells was shown by Dierks and colleagues. Using a mouse model of B cell lymphoma and human transformed cell lines, they showed that malignant B cells upregulate BCL-2 following activation of the Hedgehog pathway. Notably, this activation was mediated by the Hedgehog ligands Ihh and Shh secreted by stromal cells, and inhibition of this interaction lead to reduction of lymphoma survival and expansion in vivo [14]. Since then, other stromal soluble factors, such as IL-6, were identified as being associated with the upregulation of BCL-2 and MCL1 by BMSCs [15,16,17], highlighting the importance of these proteins to protect leukemic cells from apoptosis. However, the anti-apoptotic signals originating from BMSCs are indisputably more complex. In order to understand the genetic basis for this anti-apoptotic response, we had assessed gene expression changes induced in primary Chronic Lymphocytic Leukaemia (CLL) cells by direct contact to BMSCs. Though controlling this experiment is difficult since primary cells in culture are prone to die in the absence of stromal cells, we identified hundreds of genes activated in response to cell contact, with functions in signalling, cell adhesion and metabolic pathways [18]. Alignment of these data to the gene expression data from primary ‘bone marrow-derived’ CLL cells, published by the Wiestner group [19], identified a 31% overlap between their and our data (unpublished comparison), indicating that co-culture in vitro systems using BMSCs and primary tumour cells sufficiently recapitulate important cell–cell interactions found in patients. Importantly, the majority of concordantly regulated genes from this analysis were enriched in the gene sets ‘Purine Metabolism’, ‘lipids and lipoproteins’ ‘Glucose metabolism’ and ‘PI3K/AKT activation’, strongly indicating that stromal cells have fundamental effects on signalling and metabolic pathways in CLL cells.

### 3.2. BMSCs and Oncogenes

Interestingly, this analysis also revealed that stromal cells can directly stimulate the activity of well-known oncogenic pathways (Figure 1). In brief, it was shown that the central regulator of WNT signalling, β -catenin, was stabilised by direct cell contact to BMSCs. The underlying mechanisms involve phosphorylation of GSK3-β and transcriptional up-regulation of N-Cadherin, thereby allowing β-catenin to accumulate and to co-activate transcription in the nucleus [18]. Notably similar results were obtained by other groups in acute lymphoblastic leukaemia, demonstrating that microenvironment-mediated stabilisation of β-catenin also increases drug resistance [20,21].

The activity of the stromal compartment to stimulate oncogenic pathways in leukemic cells is not limited to WNT as recent data show that BMSCs are also critical for stimulating NOTCH signalling. Mammalian cells possess four different NOTCH single pass transmembrane receptors, which are activated by cell–cell contact through specific ligand interaction with the extracellular domain of NOTCH proteins. The activation of the receptor triggers a proteolytic cleavage of the protein that results in the subsequent release in the cytoplasm of the NOTCH intracellular domain (NICD) from the membrane of the cell. NICD translocates into the nucleus where it then interacts with other co-transcription factors to initiate transcription of NOTCH target genes [22]. Cumulative evidence indicates that NOTCH1 plays a role in tumour development and progression of different B cell malignancies [23]. Commonly, NOTCH signalling is aberrantly activated by mutations [24]; other studies in Multiple Myeloma (MM) and CLL show that this signalling pathway can also be stimulated by cancer cell adhesion to BMSCs, leading to proliferation and protection to drug-induced apoptosis [25,26].

Cell–cell contact of leukemic cells with BMSCs can also redirect the dependence of malignant cells to specific pathways. One such signalling network involved in the pathogenesis of many B cell malignancies is modulated by the transcription factor STAT3 [27,28] Multiple Myeloma cells are known to be highly dependent on ‘STAT3-mediated’ IL-6 signalling for survival in standard cell culture conditions. However, co-culture with BMSCs renders MM cells independent of IL-6R/STAT3 signalling, instead activating MEK1/2 and ERK1/2 due to induction of RAS signalling [29].

### 3.3. BMSCs and Metabolism

The metabolism of cancer cells differs largely to that of normal cells [30]. As tumour cells are normally associated with increased proliferation and an ability to survive for a prolonged time, their biosynthetic demand is massively augmented. In order to fulfil this requirement, cells must increase the import of nutrients. Therefore, it is not entirely surprising that pro-survival activities of BMSCs are associated with metabolic changes (Figure 1). The ability of leukemic cells to take up critical nutrients and metabolic products from the tumour niche is indeed critical for their survival and expansion. This is exemplified in a study by Marlein and colleagues, showing that the BM niche supports MM cells proliferation by promoting mitochondrial-based oxidative phosphorylation and glycolysis and that this support is mediated by transfer of mitochondria through tumour-derived tunnelling nanotubes [31]. Similarly, a recent study demonstrated that the presence of pyruvate released by isolated CAF from patients with lymphoma, increases the availability of the intermediates of the citric acid cycle, inducing tumour cells to produce energy by aerobic metabolism and increasing their viability in vitro [32].

## 4. Reciprocal Effects On BMSCs

The interaction between malignant B cells and BMSCs is a dynamic process driven by bi-directional communications between the two, as bone marrow stromal cells can be activated and remodelled by leukemic cells and vice versa [33] (Figure 2). This remodelling creates a de novo gene expression signature and cytoskeletal modifications in the stromal compartment that has close similarities with the features described for cancer associated fibroblasts (CAFs) present in solid tumours. Distinctive characteristics of these cells include the upregulation of pro-inflammatory cytokines, the over-expression of activated markers, such as alpha-smooth muscle actin (α-SMA), calponin and vimentin on the surface and cytoskeleton remodelling with the expression of stress fibres [34]. The importance of this cell “transformation” is not only to provide physical support for the tumour cells but also plays a direct role in promoting tumorigenesis through the modulation of the immune system (immune evasion). Even though many studies have highlighted the importance of reprogrammed stromal cells for tumorigenesis (reviewed in [35]), their in vivo presence remains poorly defined due to a lack of specific markers. Because of the wide-range effects of stroma cells on tumour cells, it is of paramount importance to understand the molecular mechanisms underlying this cell-to-cell communication as it may open new possibilities for treatment of B cell malignancies. Here we discuss the main conduits used by leukemic B cells to reprogram BMSCs.

### 4.1. Cell Adhesion Mediated Remodelling of Stroma Cells

Adhesion of malignant cells to the extracellular matrix and neighbouring cells of the tumour microenvironment (TME) is a main pillar of cell communication [36]. These adhesive interactions alone are able to trigger the activation of pro-survival and proliferative pathways in tumour cells. Integrins and chemokines are the major groups of proteins involved in cell-surface-adhesion. One of the best characterised molecular interactors is the vascular cell adhesion molecule 1 (VCAM-1)/very late antigen-4 (VLA-4) signalling [37,38] and the CXCR4/CXCL12 axis [39]. VCAM-1/VLA-4 communication plays a key role in stroma remodelling in numerous B cell malignancies: in B cell Acute Lymphoblastic Leukaemia (ALL), Jacamo and colleagues demonstrated that following VCAM-1/VLA-4 interaction, BMSCs show increased NF-κB signalling activation [38]. Subsequently, this activation drives the expression of pro-inflammatory genes, creating a stromal dependent pro-tumorigenic microenvironment. Notably, this creation of an inflammatory niche can be found in different B malignancies. In Multiple Myeloma cells, adhesion to stromal cells through integrin β1, induce the release of IL-6 from BMSCs, most likely as a result of induction of NF-κB signalling [40,41]. Of clinical importance is the increased osteoclastic activity through VCAM-1/VLA-4 contact induced by myeloma cells [42], likely contributing to bone lesions. The CXCR4/CXCL12 axis is crucial for leukemic cells homing to the bone marrow in order to create niches where malignant cells can be protected from apoptosis [43].

Recently, a new type of cellular communication has been described based on direct cell contact of long actin-rich membrane extensions named tunnelling, or membrane nanotubes (TNTs) [44]. Cellular protrusions help to transport several types of cargo, containing signalling molecules such as vesicles, miRNAs and proteins between cells. As most of the bystander communication between BMSCs and malignant B cells is mediated by close proximity, it is possible that TNTs also play a fundamental role in the reprogramming of the TME. An example of this has been shown by the work of Polak and colleagues, which demonstrates that ALL cells signal to BMSCs through TNTs, triggering the secretion of pro-survival cytokines, such as IL-8 and interferon-γ–inducible protein 10/CXC chemokine ligand 10 (CXCL10) [45].

### 4.2. Soluble Determinants of Stroma Remodelling

The second pillar of cell-to-cell communication constitutes of soluble factors, including cytokines, chemokines and growth factors. Cytokines and growth factors play a major role in regulating a wide range of biological functions in both physiological and pathological conditions, including immunity processes, haematopoiesis, inflammation, cell proliferation and differentiation. Their secretion is tightly controlled and, within the TME, they extend their function in a predominantly paracrine fashion to closely adjacent cells [46]. As cytokine and growth factor production of leukemic cells differ from those of PBMC and normal cells [47], they play an essential role in tumour microenvironment reprogramming. In CLL, Ding et al. showed that platelet-derived growth factor (PDGF) can activate BMSCs, enhancing their proliferation and inducing PI3K signalling. This activation results in increased VEGF secretion, shown to protect primary CLL cells from apoptosis [48,49]. In MM, the tumour necrosis factor alpha (TNFa) plays a major role: once it is released by malignant cells, NF-kB signalling in BMSCs is activated, resulting in the upregulation of IL-6 secretion [50]. Importantly, this is associated with reduced osteogenic potential of bone marrow stromal cells, ultimately resulting in decreased bone regeneration [51].

Similarly, the ability to influence the lineage differentiation of stromal cells exhibited by leukemic cells through release of TNFα is also evident in the lymph node (LN). Similar to the BM, human secondary lymphoid organs (SLO) also contain stromal cells of mesenchymal origin [52]. Similar to BMSCs, these cells are also able to sustain the survival of leukemic B cells and play a role in the proliferation and progression of the tumour. In Follicular Lymphoma (FL), release of TNF-alpha and lymphotoxin alpha1beta2 (LTA1B2) by leukemic cells triggers the SLO stromal compartment to differentiate towards the fibroblastic reticular cell (FCR) lineage that exhibit an improved support for the malignant B cell survival, potentially mimicking the CAF-like reprogramming observed in BMSCs [53].

Different studies have shown that in addition to releasing cytokines/chemokines, leukemic cells secrete vesicles, allowing for the communication between cells, in some cases even without the need for specific receptors on the stromal counterpart [54]. Exosomes are a particular class of vesicles and, contrarily to microvesicles, originate from the endocytic pathway rather than form the plasma membrane. They are nanometre sized (normally 30–100nm) and can contain proteins, DNA, coding and noncoding RNAs, able to influence the functions and the signalling pathways of target cells. In CLL, Paggetti et al. show that exosomes originated from leukemic cells are incorporated by BMSCs in vivo and in vitro and promote stromal cell activation, leading to enhanced proliferation, migration and secretion of pro-inflammatory cytokines mediated by NF-kB [55]. The gene signature and the cellular phenotype of these activated BMSCs show increased levels of tumour necrosis factor family ligands, pro-inflammatory cytokines (e.g., IL-6), chemokines (e.g., CCL2, CXCL16), and proangiogenic factors (e.g., HGF), resembling the phenotype of CAFs found in solid tumours. Interestingly, they found that particular miRNAs, such as miR-21 and miR-146 that are known to be associated with stromal remodelling, are enriched in exosomes released by leukemic cells. Furthermore, the effects of the exosomes extended to the endothelial compartment, leading to enhanced angiogenesis [55]. Similar results were also obtained with ALL cells. Johnson et al. demonstrate that BMSCs not only incorporate exosomes, originated from ALL cells, and reprogram them into CAFs like cells, but also that these cells undergo a metabolic switch following exosome uptake. This switch induced BMSCs to release more lactate into the supernatant that could be used by the cancer cells as an alternative source of energy [56]. Tumour derived exosomes are also fundamental in Multiple Myeloma (MM) where BMSCs take up MM-derived vesicles, mainly mediated by endocytosis, and this activates pro-survival and pro-proliferation pathways [57].

### 4.3. Activated Signalling Pathways in Remodelled BMSCs

As described above, the cell-to-cell communications between malignant B cells and stromal cells are based on numerous membrane-bound mechanisms and soluble factors. While a plethora of these factors have been identified in numerous B cell malignancies, little is known about the following down-stream molecular events in stroma cells. Unambiguously, most evidence exists for the activation of NF-kB, of key importance for driving cancer-induced inflammation. Briefly, the NF-kB family is composed of five related transcription factors: p50, p52, RelA (p65), c-Rel and RelB that are the master regulators of inflammatory processes [58]. The two main signalling pathways that lead to NF-kB activation are known as the canonical and non-canonical pathway, mainly differing from their dependence on the IkB kinase adaptor molecule NEMO and IKKβ (IKK2), both required for the canonical activation of NF-kB, while the non-canonical pathway depends on IKKα (also called IKK1). Phosphorylation and subsequent degradation of IkBα leads to the release of p50 and RelA and canonical activation of NF-kB, while this is achieved through phosphorylation and processing of p100 into p52 and RelB [59]. The NF-kB canonical pathway is generally associated with an inflammatory response, whereas activation of the non-canonical pathway is mostly related to developmental cues [60]. NF-kB is thought to be the dominant driver for the production of different cytokines and adhesion molecules that help leukemic cells to survive and proliferate [61]. The importance of this pathway is undeniably enlightened by the fact that it is essential for the typical role of BMSCs to sustain the survival of primary malignant B cells in vitro. As mentioned in the previous paragraph, different pathways are involved in NF-kB activation. Data from our group identified stroma PKC-β as one of the main factors involved in the regulation of this pathway following cell–cell contact. This kinase, when induced and activated, operates upstream of canonical NF-kB activation and is an essential prerequisite for NF-kB activation in stroma cells. This stimulation then results in enforced expression of adhesion molecules and pro-inflammatory cytokines, such as IL-1 and IL-15, synergistically blocking apoptosis of leukemic cells. The importance of this pathway in the TME was also demonstrated in vivo, as mice carrying a germline deletion of PKC-β were refractory to CLL-like transplantations [62].

NOTCH signalling is another important pathway regulated by leukemic cells in BMSCs. NOTCH signalling is mainly associated with cell differentiation cues and mutations in pathways that lead to disease formation or progression in various organs. Not much is known about its role in BMSCs in physiological conditions; some studies suggest that it can play a role in endothelial differentiation [63], to initiate chondrogenesis [64] and to maintain the pool of mesenchymal progenitors in the bone marrow [65]. However, in pathological conditions, different studies have shown that leukemic cells not only activate NOTCH signalling [66] but also induce NOTCH signalling in stroma cells. In ALL, leukemic cells express higher levels of Jagged1/2 compared to normal cells. Indeed, co-culture of ALL cells with BMSCs leads to aberrant activation of NOTCH signalling with the consequence of decreased expression of osteogenic markers that can ultimately result in the remodelling of the bone marrow microenvironment in vivo. ALL patients with osteolytic bone lesions or osteoporosis, in fact, show higher expression of Jagged1 compared to patients lacking these clinical features [67]. Similarly, MM cells remodel the bone marrow microenvironment by regulating osteoblast and osteoclast activity through NOTCH signalling. MM cell-mediated activation of NOTCH is capable of reducing the viability of osteocytes due to increased apoptosis [68]. Furthermore, Colombo et al. show that MM cells activate NOTCH signalling in BMSCs through expression of Jagged1/2, which results in the production of growth factors such as IL-6, VEGF and IGF1. All of them have beneficial effects for the survival of malignant cells [69,70]. Similar to myeloma cells, CLL cells also express NOTCH ligands [18,26,66]. Recent data from our group have shown that, in a contact dependent manner, leukemic cells specifically activate NOTCH2 signalling in BMSCs with an associated loss of NOTCH1. This activation leads to the production of soluble factors, such as C1q, which has been shown to promote CLL cell survival, similar to effects observed on metastatic lesions of solid tumours [18,71]

### 4.4. Metabolic Remodelling of BMSCs

The bystander relationship between cancer cells and BMSCs is not only limited to secretion of cytokines and other stromal factors but is also associated with cell metabolic reprogramming of the latter, which contributes to cancer growth and progression. The bone marrow microenvironment is highly hypoxic and hypoglycaemic [72], and it is, therefore, not surprising that leukemic cells can modify metabolic pathways of neighbouring cells in order to satisfy their high energy demands. In CLL for instance, cells have limited ability to take up cystine, the oxidised dimer form of the amino acid cysteine, necessary for the maintenance of the intracellular glutathione (GSH) level, due to low levels of the Xc-transporter. Contrary, BMSCs instead express high levels of Xc- transporter and this allows them to take up cystine. Cystine is then converted to cysteine and released into the microenvironment, available for tumour cells. Eventually, this metabolic remodelling is necessary for tumour cell viability and drug resistance [73]. Similarly, ALL cells through the release of insulin-like growth factor-binding protein 7 (IGFBP7) induce insulin/insulin-like growth factor (IGF-1) signalling in BMSCs. This activation results in enhanced asparagine synthetase expression and, consequently, more asparagine secretion that can be used by ALL cells for their metabolic needs, causing a relative resistance of ALL cells to asparaginase [74]. Leukemic cells also induce expression of a protein with enzymatic activity and metabolic transporter in stromal cells. For instance, interfering with the stromal expression of LDHB and PKM1 revert the lactate acidosis seen in some patients with aggressive B cell lymphoma [75]. This is defined as the reverse Warburg effect that proposes that tumour cells can provoke BMSCs to become glycolytic and export lactate, which is then taken up by cancer cells and used for oxidative metabolism. The metabolic characterisation of the tumour microenvironment is still in its infancy, and it can be expected that this field will further grow in the near future with improved technologies and better in vivo and in vitro tumour models.

## 5. Conclusions

A growing amount of evidences indicate that the interaction between stroma and malignant B cells regulates a broad range of biological processes, which are of key importance for disease progression. This once more underscores that tumours are complex organoids, composed of a variety of non-malignant and malignant cells. These cell–cell interactions are dynamic and provide the basis for the plasticity of cancer. While, as we outlined in this review, numerous key findings were obtained from somehow “crude” in vitro experiments, what needs to follow in the near future is a validation of these observations in vivo, in appropriate disease models and also in patients. As illustrated by the recent publications [7,9], single-cell technologies will be a major advantage to address this.

We believe that a meticulous functional characterisation of the tumour-stroma compartment will open new possibilities for treatments. Since malignant B cells heavily rely on receiving nurturing factors from stromal cells in order to survive and proliferate, interference with this dependency is likely to be of therapeutic value. We speculate that such therapeutic approaches may be superior to therapies targeting genes in tumour cells: contrary to the latter, no evidence exist that stromal cells are genetically unstable [76]; hence, clonal selection for “resistant” stroma cells is not expected to happen. Ultimately, combined treatments, simultaneously targeting stroma and tumour cells, are likely to be synergistic and to result in superior effects compared to treatments focussing on killing tumour cells only.

## Figures and Tables

**Figure 1 ijms-21-01466-f001:**
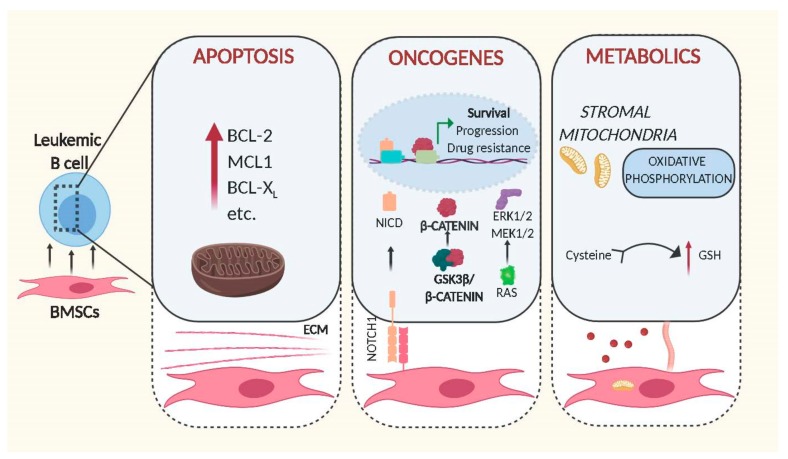
Schematic representation of intracellular changes mediated by bone marrow stromal cells in leukemic B cells. Cell–cell contact and release of soluble factors mediated by BMSCs increase the leukemogenicity of transformed B cells through expression of anti-apoptotic proteins (left panel), activation of oncogenic signalling (middle) and alteration of metabolic pathways (right). Figures were generated using Biorender.com.

**Figure 2 ijms-21-01466-f002:**
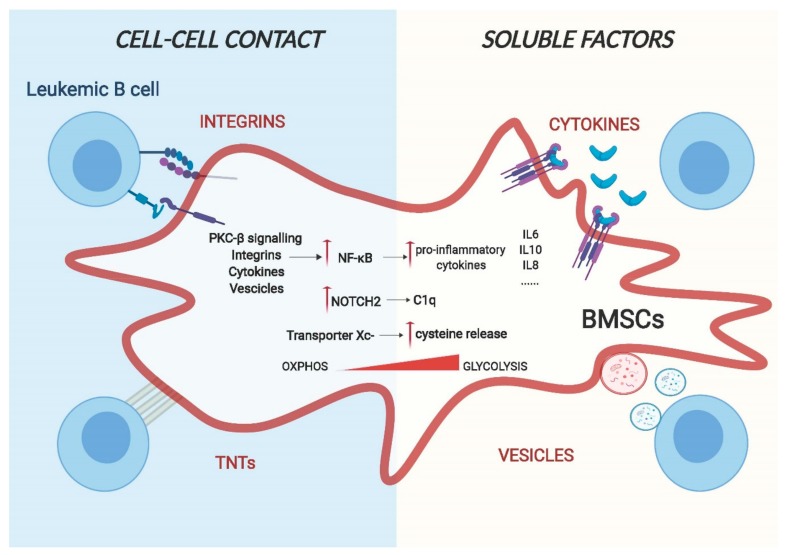
A model for cell–cell interaction between leukemic B cells and bone marrow stromal cells. Cell adhesion and release of soluble factors in the microenvironment are the two main routes for cell–cell communication. Leukemic cells can change the gene expression and the metabolic behaviour of BMSCs through transmembrane proteins, such as integrins and chemokines, or through using actin-rich membrane extensions called tunnelling nanotubes (TNTs) via a direct physical interaction. Additionally, they can also release soluble factors such as cytokines and vesicles that are capable of regulating the phenotype of stromal cells.

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
