# Peer review of "Bone Marrow Stromal Cells Drive Key Hallmarks of B Cell Malignancies"

_ijms, 2020, doi:10.3390/ijms21041466_

Round 1

Reviewer 1 Report

Your paper extensively reports a list of observations and data concerning the complex interaction between lymphoid malignant cells and bone marrow stromal cells. Although the topic is very interesting, I find some critical issues. Namely, the term "lymphomagenesis" refers to events not addressed in the review. It is unclear the link between MRD and lymphomagnesis. Referring to tumor cells, very different lines of hematological malignancies are considered, also myeloid neoplasms.  You talk about  of lymphomas as indolent diseases. Lymphoma are various and present different degree of response to therapies. So I suggest you to review the points I have raised.    

Author Response

We thanks this reviewer for pointing out that the term “lymphomagenesis” may be misleading. (We intended to use it to describe the processes required for maintenance and progression of B cell malignancies, but we acknowledge that in a strict meaning biological processed leading to malignant transformation can be meant). We have therefore decided to eliminate this term from the manuscript and also have changed the title of the manuscript accordingly.

We have summarized and reviewed evidences of stroma to B cell effects. We believe this is an appropriate approach since similar effects have been observed e.g. between cells from ALL patients and CLL patients (aggressive and indolent). This may simply reflect a cell of origin dependency. We have carefully indicated what type of B cell malignancy we are referring to.

We have only referenced AML in context of the recent scRNA seq paper as this is the first paper investigating disease induced alterations of the stroma compartment on a single cell level. Acknowleding this reviewers criticism, we have added a sentence (page 3) following this paragraph: “However, it remains to be shown to what extent these findings apply to lymphatic malignancies”. We hope this is an acceptable solution for this reviewer.

Reviewer 2 Report

The review is well written and exhaustive

Author Response

We appreciate the positive feed back from this reviewer.

Reviewer 3 Report

OVERVIEW

The authors present a very interesting and timely review of the microenviromental molecular factors involved in B-cell leukemia and lymphoma. The authors have contributed to this topic and the review covers the latest findings in a concise and clear manner, citing the main publications. Although a few revisions are required (most are language revisions), in general the manuscript is clearly of value for researchers and clinicians interested in the latest developments in hematological malignancies.  

MAJOR COMMENTS

In page 2, line 34, the abbreviation ‘BMSCs’ is introduced to mean ‘bone marrow MSCs’. However, throughout the text, and consulting the cited references, it becomes apparent that the abbreviation ‘BMSCs’ is used interchangeably to mean both MSCs (e.g. ref. 38) and stromal cells (e.g. refs 7, 10, and 11), which are not exactly the same cells. I suggest revising the text and use  BM-MSCs for ‘bone marrow-derived mesenchymal stem cells’ and BMSCs for ‘bone marrow stromal cells’. This should remove ambiguity. In addition, note that ‘BMCSs’ instead of ‘BMSCs’ appears 4 times in the text (pages 7 and 8). In page 7 the differences between the two NF-kappaB pathways is not very clear. In line 9, it should mentioned that the canonical pathway depends on NEMO (correct) and IKKbeta (also called IKK2), while the noncanonical pathway depends on IKKalpha (also called IKK1) and NIK kinases. In addition, in line 12, it should be mentioned that the canonical pathway activation involves IkappaB phosphorylation and degradation, while the noncanonical pathway is activated by p100 phosphorylation and processing to p52. In page 7, line 33, there is a discrepancy because the text refers to ALL while ref. 66 refers to B-CLL. Moreover, ref. 66 could perhaps be cited in line 44, together with ref. 18, and replace Ref. 71, which is the same as ref. 26. In page 8, line 12, the description of the findings from ref. 75 are not very accurate. The paper reports in vitro co-culture experiments, so extrapolating to TME (tumor microenviroment) would require in vivo experiments that they did not do. In addition, the main conclusion that Laranjeira et al, reached was that increased asparagine secretion improved resistance to Asparaginase treatment. It was not clarified that asparagine was important for steady-state metabolic needs, in vitro or in vivo. Figure 1 and 2 should be cited in the main text.

MINOR REVISIONS

In the lines 3-4 of the Abstract the word “side” should be “site”. In line 6 of Introduction, the term ‘clonal and sub-clonal events’ should be precised. Does it mean ‘ genetic mutations’, ‘epigenetic alterations’ or ‘differential gene/protein expression’. In line 11 of Introduction, ‘BH3 proteins’ should be replace by ‘BH3-only proapoptotic proteins’. In line 13 of Introduction, does ‘remains to date’ suppose to mean ‘remains up-to-date’? Perhaps the sentence needs to be more clear. In page 2, line 7, which procedure should be clarified. Is it bone marrow aspirate? In page 2, line 8, MDR should be MRD. In page 2, line 10, what is meant with “spatial requirements” could be more clear. I guess the authors mean tissue, organ or niche requirements, but it is not straightforward. In the same sentence, there should be a comma after ‘described’. The authors wrote the text in UK English, so many words should be converted from US to UK English: ‘hematopoietic’, ‘leukemic’, ‘tumorigenesis’, ‘signaling’, ‘tumors’, and ‘leukemogenicity’ Several words should be hyphenated: ‘stem-like’ (page 2), ‘haematopoietic-supporting (page 2), ‘drug-induced’ (page 3), ‘bone marrow-derived’ (page 4), and ‘STAT3-mediated’ (page 4). In page 3, section title, BMSCs should be in capital letters. In page 4, line 4, there should be a comma after ‘then’. The nomenclature for interleukins should be uniform, choosing between ‘IL6’ or ‘IL-6’. The same for other interleukins mentioned in the text. The verb ‘to uptake’ (page 5, line 2; page 8, line 4) does not exist. It should be replaced by ‘to take up’. In page 5, line 3, ‘survive’ should be corrected to ‘survival’. In page 6, it would be more appropriate to designate the lymphotoxin heterotrimers as ‘lymphotoxin alpha1beta2’ instead of ‘LTA1B2’. In page 7, line 20, write ‘... This kinase, when activated and induced,...’ In page 8, line 9, it appears that the word ‘resistance’ after ‘drug’ is missing. In the Figure 2 schematic, ‘vesicles’ is misspelled to ‘vescicles’.

MINOR SUGGESTIONS

In page 2, line 33, use the word ‘drive’ instead of ‘eagerness’. In page 8, line 26, ‘findings were obtained’ seems more correct than ‘findings were made’ In page 8, line 30, suggestion of writing ‘meticulous functional characterization’. In page 8, line 33, suggestion of writing ‘to be of therapeutic value’.

Author Response

We thank this reviewer for their careful evaluation. We have made changes to the manuscript as suggested in major and minor revisions and suggestions. With regard to the use of UK English, we prefer to keep UK English as the this is a European Journal (though this might sounds cynical in light of Brexit) and all authors are UK based.

We appreciate the comment and clarification regarding the terminology of MSC and BMSC. We have adopted the term BM-MCs for stem cells and BMSC for stroma cells as suggested by this reviewer. We hope that this facilitates reading of the manuscript. Unfortunately, the terms “stoma” and “stem” are used interchangeably by many authors. For instance, the recent paper from David Scadden’s group refers to MSC as “One major component is multipotent mesenchymal stem/stromal cells (MSCs), non-hematopoietic cells derived from the mesoderm” Baryawno et al., 2019, Cell 177, 1915–1932”.

We thank the reviewer for their clarification and added more information regarding the canonical and non-canonical NFkB signalling pathways as suggested.

Reference 66 was misplaced in the text- this has now been corrected. The thank the reviewer to have spotted that there was a duplication of a reference. This has been changed according to this reviewers suggestions. We have added information to clarify that the Laranjeira paper was not investigating the TME in vivo and that resistance was with reference to asparaginase.

Figures 1+2 are now mentioned in the text.

We hope that the manuscript in its revised form is now acceptable for this reviewer.

Round 2

Reviewer 1 Report

I appreciate the changes made in relation to my suggestions.

However, I would like to correct the sentence of "dogma of indolent lymphomas".

Only a fraction of B-lymphoma have an indolent behaviour.  I'd talk about indolent lymphoma and relapse/resistance of high-grade lymphomas.

In order to be more precise, I suggest you one sentence more fitting